 PLOS Biology

# Biased sampling driven by bacterial population structure confounds machine learning prediction of antimicrobial resistance

Yanying Yu[1,2], Nicole E. Wheeler[3], Lars Barquist[1,4,5]*

1 Helmholtz Institute for RNA-based Infection Research (HIRI), Helmholtz Centre for Infection Research (HZI), Würzburg, Germany, 2 Division of Infectious Diseases, Boston Children's Hospital and Department of Pediatrics, Harvard Medical School, Boston, Massachusetts, United States of America, 3 Institute of Microbiology and Infection, University of Birmingham, Birmingham, United Kingdom, 4 Department of Biology, University of Toronto, Mississauga, Canada, 5 Department of Cell and Systems Biology, University of Toronto, Toronto, Canada

* lars.barquist@utoronto.ca

## Abstract

Antimicrobial resistance (AMR) poses a growing threat to human health. Increasingly, genome sequencing is being applied for the surveillance of bacterial pathogens, producing a wealth of data to train machine learning (ML) applications to predict AMR and identify resistance determinants. However, bacterial populations are highly structured, and sampling is biased towards human disease isolates, violating ML assumptions of independence between samples. This is rarely considered in applications of ML to AMR. Here, we demonstrate the confounding effects of sample structure by analyzing over 24,000 whole genome sequences and AMR phenotypes from five diverse pathogens, using pathological training data where resistance is confounded with phylogeny. We show the resulting ML models perform poorly and that increasing the training sample size fails to rescue performance. A comprehensive analysis of 6,740 models identifies species- and drug-specific effects on model accuracy. These findings highlight the limitations of current ML approaches in the face of realistic sampling biases and underscore the need for population structure-aware methods and more diverse datasets to improve AMR prediction and surveillance.

## Introduction

Antimicrobial resistance (AMR) is a severe threat. Currently, an estimated 4.95 million deaths are associated with AMR each year [1], and this number is only predicted to increase. Urgent action is required to prevent further escalation, slow epidemic spread, and preserve our capacity to administer advanced medical therapies relying on surgery or immunosuppression. Surveillance based on whole genome sequencing is playing an increasingly important role in monitoring the emergence and spread of AMR [2]. Taking full advantage of this accumulating data will require automated

**Data availability statement:** All code necessary to reproduce the results in the manuscript is available at https://github.com/BarquistLab/AMR_prediction (archived at Zenodo, DOI: https://doi.org/10.5281/zenodo.17399563). Intermediate files and results of machine learning have been deposited at Mendeley under DOI: https://doi.org/10.17632/zs2mbjv7dn.3.

**Funding:** This work was funded in part by the Bavarian State Ministry for Science and the Arts through the research network Bayresq.net (to L.B.), and an Natural Sciences and Engineering Research Council (NSERC, https://www.nserc-crsng.gc.ca/index_eng.asp) Discovery Grant (RGPIN-2024-04305 to L.B.). The funders played no role in study design, data collection and analysis, decision to publish, or preparation of the manuscript.

**Competing interests:** The authors have declared that no competing interests exist.

**Abbreviations:** AMR, antimicrobial resistance; AUC, area under the ROC curve; GWAS, genome-wide association studies; LMICs, low- and middle-income countries; ML, machine learning; SHAP, Shapley additive explanation.

systems that can transform genome sequences into actionable intelligence, namely, predictions of resistance profiles and identification of new mechanisms of resistance [3–5]. Machine learning (ML) approaches have the potential to fill this niche, though a number of challenges remain in implementing such systems [6].

One underappreciated challenge is in the assumptions made by ML methods themselves. Most classical ML methods assume training data are independent and identically distributed, which is not true of pathogen surveillance samples due to the underlying structure of bacterial populations [7]. During the epidemic spread, successful clones spread rapidly. If this spread is due in part to acquisition of AMR determinants it could lead to an association between the phenotype and phylogenetic markers that do not directly contribute to AMR. These noncausative associations are likely further exacerbated by biased sampling focused on human disease in high-income countries [5,8] leaving large regions of the phylogeny unexplored. The phylogenetic dependence of genes and traits has long been recognized in comparative and evolutionary biology [9], with statistical approaches to correct for this dating at least to the mid-1970's [10]. In the context of microbial genome-wide association studies (GWAS), these phylogenetic effects have been mitigated in part by the application of mixed-effect models that attempt to correct for sampling biases [11–13]. To date, the implications of biased sampling for ML approaches to AMR prediction have not been systematically investigated, despite the clear risk of confounding [14]. Here, by constructing realistic pathological scenarios, we show that biased sampling combined with population structure can dramatically affect the performance of ML methods for predicting AMR. We investigate the factors underlying the performance of ML methods and provide concrete recommendations for the future evaluation of ML approaches to predicting AMR.

## Sampling effects bias machine learning prediction of AMR

To comprehensively evaluate the impact of population structure in predicting AMR, we collected between 3,204 and 7,188 genomes for each of three Gram-negative species and two Gram-positive species representative of current WHO priority pathogens [15], including the gastrointestinal and urinary tract pathogen *Escherichia coli,* the opportunistic pathogen *Klebsiella pneumoniae,* the gastrointestinal pathogen *Salmonella enterica*, the skin commensal and opportunistic pathogen *Staphylococcus aureus*, and the major agent of community-acquired pneumonia *Streptococcus pneumoniae*. In total, our dataset includes resistance phenotypes for 27 antibiotics, spanning multiple drug classes and diverse sequence types (S1–S6 Figs, Methods). To limit the effects of sample size and class imbalance, we excluded antibiotic-organism combinations with less than 1,000 genomes or with resistant or susceptible strains exceeding 80% of the data set. The average number of genomes was ~2,700 with ~44% resistant strains, and 80% of the data was in the range of 1,134–3955 genomes, with 25%–71% resistant. A whole genome alignment was constructed for each species using Roary [16], and SNP-sites [17] was used to identify variable positions in the alignment. Both SNPs and the pangenome presence-absence matrix from Roary were used as predictive features. As a representative ML model we used

LightGBM [18], a gradient boosted decision tree method representative of the tree ensemble-based methods that generally perform well for predicting AMR [19,20].

We defined discrete clades based on deep divergences between branches observed in the phylogenetic tree for each species constructed from core genome alignments using IQ-Tree (Figs 1A, 1B and S2–S6A) based on deep divergences between branches. These clades align closely with established sequence types, providing biologically meaningful groupings. Using this clade structure, we developed pathological training cases simulating biased sampling. In our first approach, Scheme A (Fig 1C), we selected two clades, and a balanced test set was held-out from one (the test clade). We then devised 4 scenarios, representing biased sampling. Scenarios a and b excluded either resistant or sensitive isolates from training data for the test clade, respectively. As a positive control, we also included scenarios c and d, which excluded sensitive or resistant strains from the nontest clade. Compared to a random split of training and test data, models trained on biased datasets exhibited much lower area under the receiver operating characteristic curve (AUC) scores (Fig 1D). Exclusion of resistant strains led to low precision and high recall of sensitive strains (S7A Fig), while the opposite was observed for exclusion of sensitive strains (S7A Fig), suggesting that the ML models may have been influenced by

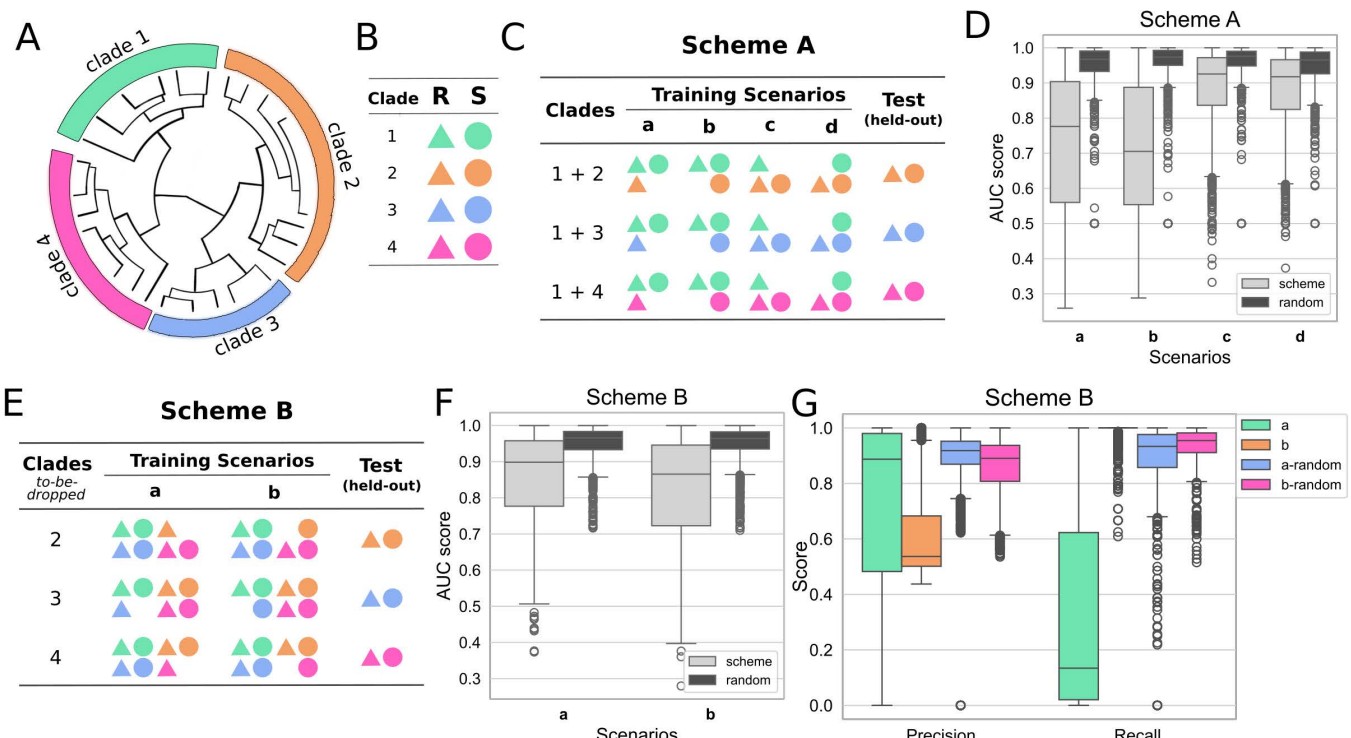

**Fig 1. Bacterial population structure and clade-specific sampling affect machine learning predictions of antimicrobial resistance (AMR).** We define clades based on core genome phylogeny and simulate training and testing scenarios that highlight challenges in generalizing AMR prediction across genetically distinct groups. (A) An illustration of defining clades for genomes in one species. (B) Legend for panels C and E, R: resistant (triangles), S: susceptible (circles). (C) An example of training scenarios for scheme A based on the clade splitting in A. Models were trained on clade 1 and one of the other clades, such as clades 1 and 2, clades 1 and 3, and clades 1 and 4. Four training scenarios were tested for each antibiotic and species. In each scenario, either resistant or susceptible samples from one of the paired clades were excluded from the training data, while the held-out genomes from the clade that is not clade 1 were used as the test sets. (D) The area under the ROC curve (AUC) scores in each scenario for scheme A for all antibiotics and species. Light gray: train-test split followed scheme A, dark gray: random train-test split. (E) An example of training scenarios for scheme B based on the clade splitting in A. Models were trained on all samples except for resistant or susceptible samples from one clade. The held-out genomes from the same clade were used as the test sets. (F) The AUC scores for each scenario for scheme B for all antibiotics and species. Light gray: train-test split followed scheme B, dark gray: random train-test split. (G) The precision and recall scores for scheme B; a-random and b-random indicate testing on random train-test splits of the pathological unbalanced data. Underlying data for this figure are available in S3 Data.

markers of lineage rather than identifying genuine indicators of AMR. This was not observed in our control scenarios, and, importantly, persisted even when both resistant and sensitive strains were present in the test clade, but at biased proportions (S7BC Fig).

It is often implicitly assumed that simply increasing sample sizes is sufficient to improve ML models. To test this, we developed a second approach, Scheme B, where we additionally included training data from remaining clades, while still excluding resistant or sensitive isolates from the test clade (Fig 1E). This did lead to an improvement in the AUC compared to Scheme A (Fig 1F). However, extreme biases remained in both precision and recall (Fig 1G), indicating that trained models continued to conflate lineage and AMR markers even in the presence of a larger training data set.

## AMR predictivity varies across antibiotics and species

To further understand the factors driving model performance, we trained a random forest meta-model to predict the AUC scores of all trained models (S2 Data), including species, training scenario, clade-wise distance, sample size, and drug class as predictors. We then interpreted this meta-model using Shapley additive explanation (SHAP) values [21] (Fig 2A). SHAP values are a game theoretic approach to model interpretation that can be intuitively understood as providing a quantitative estimate of each feature's contribution to the model prediction, in this case the AUC.

This analysis reaffirmed the inferior performance associated with our pathological training scenarios. Additionally, this analysis showed that training on data sourced from Gram-negative species yielded diminished AUC scores (Fig 2B), presumably due to the presence of the outer membrane [22], and that drug class also had large and at times contradictory effects on model performance (Fig 2C). Given the impact of the species on model performance, we asked how the model interpretation differs when training on each species (see S2–S6C Figs). As an example, the effects of fluoroquinolone varied in three species (S2C, S4C, and S5C Figs) and also in the model encompassing all species (Fig 2A). Subsequent interpretation of models trained on each clade for ciprofloxacin (S9 Fig) further underscored the distinct determinants for prediction across clades and species. Among the known genes associated with ciprofloxacin resistance [23], the gene *parC* prominently featured in most clades of *E. coli* and *S. aureus.* In contrast, the gene *gyrA* was exclusive to a single clade within *S. aureus*, possibly reflecting different fitness effects or primary target of ciprofloxacin in this clade [24]. In other clades of *E. coli* and *S. enterica*, the strongest predictors diverged, encompassing genes such as transcriptional regulator *kdgR*, mobilization gene *mobB*, quinone oxidoreductase *qorA*, and genes encoding hypothetical proteins.

To test if this lack of common features was general, we trained clade-specific models for all drug classes. Surprisingly, minimal feature overlap was identified, both within the top 10 features and when summing the top features based on SHAP values to account for 50% of total SHAP values (S8A and S8B Fig). This scarcity of common predictive genes or SNPs across clades may in part be a result of the small number of strong predictors (S8C Fig), and provides a partial explanation for the strong effects we observed for our pathological training scenarios. Together with the performance patterns observed in S7A Fig, these results support the conclusion that the models conflate lineage-associated markers with genuine indicators of AMR.

## Implications for AMR prediction

ML-based modeling of AMR can serve multiple purposes. These include at least two major applications: 1) prediction of strain resistance, and 2) identification of causal variants. While these two applications are related, achieving one does not necessarily imply achieving the other. If biased sampling reflects biological reality, i.e., all strains of a particular lineage truly are resistant to an antibiotic, then models that conflate lineage markers with resistance may still provide accurate predictions for surveillance purposes. This can be seen, for instance, in applications that have used phylogenetic placement to predict AMR [25]. However, such models will not provide insights into the underlying molecular mechanisms of resistance, and will fail on strains from previously unseen lineages or resistant strains emerging from a former sensitive

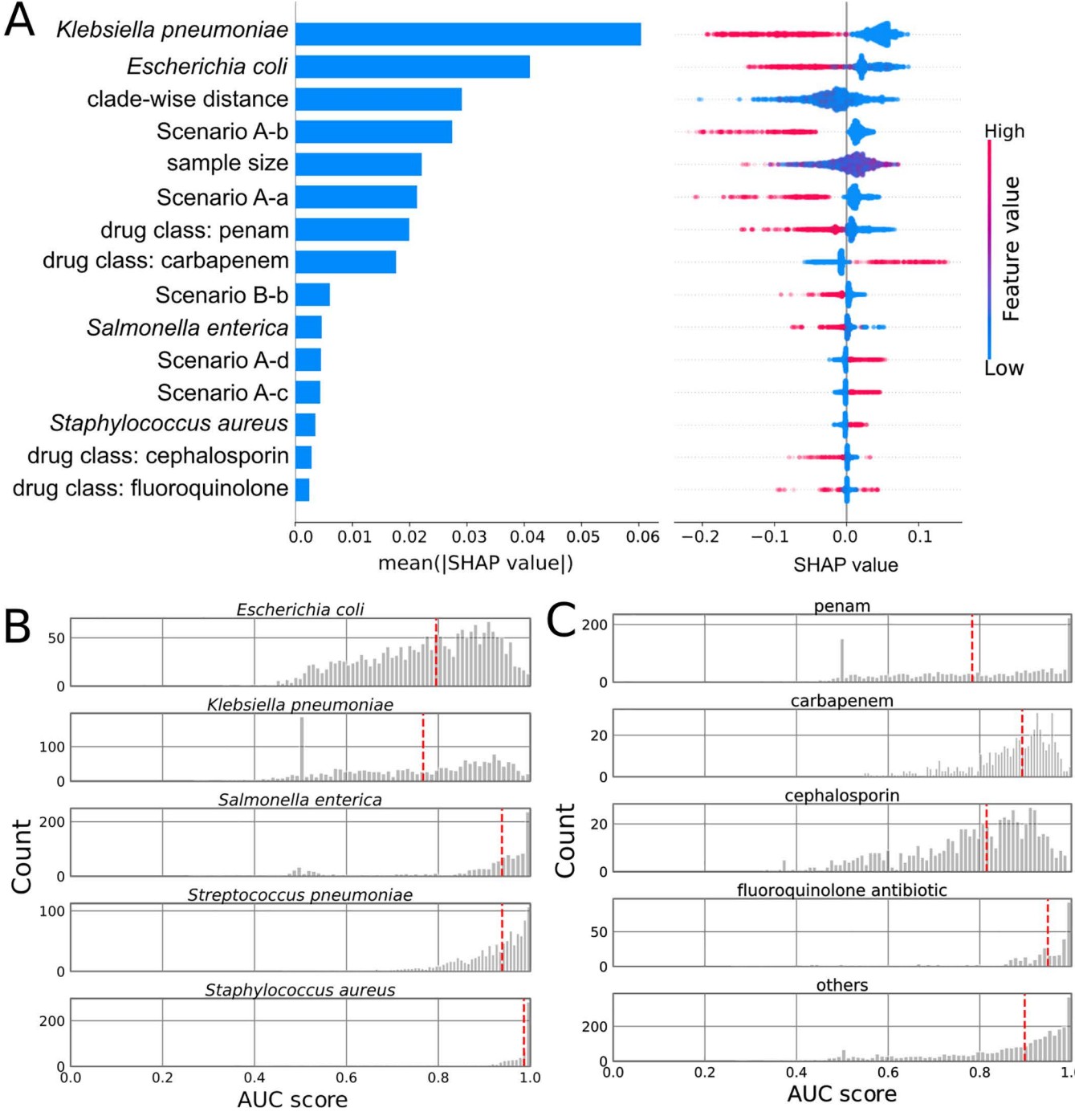

**Fig 2. AMR predictivity varies across antibiotics and species.** (A) SHAP values for the top 15 features from a random forest model trained on AUC scores from both schemes A and B for all five species. Global feature importance is given by the mean absolute SHAP value (left), while the beeswarm plot (right) illustrates feature importance for each guide prediction. (B) The distribution of AUC scores for each species. Red dashed lines indicate the median values. (C) The distribution of AUC scores for four drug classes shown in A. Red dashed lines indicate the median values. Underlying data for this figure are available in S3 Data.

lineage. Conversely, identifying causal variants requires disentangling lineage effects from true resistance determinants, a task complicated by the structure and biases present in genomic datasets.

Our results demonstrate that current ML approaches are particularly vulnerable to the confounding effects of biased sampling and population structure. Even when sample sizes are increased, models struggle to generalize beyond the specific clades included in the training set, underscoring the limitations of scaling as a solution to bias. This is supported by previous work that has shown phylogenetic structure is an important feature for predicting AMR [19], and that many AMR prediction tools perform poorly when tested on phylogenetically-constructed benchmark sets [26]. The lack of overlap we found in predictive features across clades further highlights the challenges in identifying universal resistance markers, with most predictions being dependent on the phylogenetic background. While some of these differences likely reflect true clade-specific biology, they may also be influenced by the reduced proportion of shared core genomic variation between more distantly related clades. Our ML framework, which incorporates both core genome SNPs and accessory gene presence–absence data, partially mitigates this limitation by capturing variation beyond the strict core genome. Nonetheless, this remains an important factor to consider when interpreting cross-clade predictive patterns. These findings are consistent with previous observations in bacterial GWAS, where lineage effects are a persistent source of confounding and require explicit correction.

In addition to sampling considerations, feature representation is another important factor influencing model performance. Our use of a combination of core genome SNPs and accessory gene presence–absence patterns captures broad genomic differences but does not capture sequence variation within accessory genes. This limitation is particularly relevant for species with large and diverse accessory genomes, such as *E. coli* and *K. pneumoniae*, where within-gene variation may carry additional predictive signals for AMR. To address this concern, we performed a reanalysis of a subset of our *Klebsiella* benchmark data using unitigs as predictive features, producing results consistent with our SNP and accessory gene features (see Methods, S10 Fig). While this is not a comprehensive exploration of alternative feature encodings, including *k*-mers and unitigs, it does suggest that feature engineering alone is unlikely to be sufficient to address the underlying problem of nonindependence of samples.

The statistical challenge posed by analysis of samples related by descent from a common ancestor has long been recognized [10], and our results demonstrate that ML is not immune from such concerns. Future efforts to improve ML-based AMR prediction must incorporate strategies to address this challenge. A wide range of methods have been developed in the field of comparative biology [9,27,28] that may serve as inspiration for adapting ML algorithms. One potential approach is the development of lineage-aware ML algorithms that explicitly model the hierarchical structure of bacterial populations, borrowing ideas from mixed-effect models commonly used in bacterial GWAS [11–13]. Another promising direction is the integration of population-genomic frameworks that incorporate evolutionary signals, such as measures of phylogenetic conservation, into feature selection and model interpretation [29]. Additionally, benchmark datasets that include well-curated, globally representative samples are essential for evaluating and comparing new approaches, ensuring models are tested in realistic scenarios that capture the diversity encountered in clinical and environmental surveillance. Our strategy of phylogeny-aware cross-validation, i.e., testing on held-out clades, will be an important tool to demonstrate generalizability of ML predictors of AMR.

Importantly, sampling strategies must also evolve to support these developments. Expanding surveillance efforts to underrepresented regions and settings, particularly in low- and middle-income countries (LMICs), will provide more balanced datasets and reduce biases toward high-income countries and outbreak scenarios [8]. Targeted sampling that prioritizes diversity, both within and across clades, will be critical for developing models capable of generalizing across phylogenetic backgrounds. Beyond sampling, careful experimental design, including the use of balanced test sets and rigorous evaluation metrics such as those employed here, will further ensure that models are robust to confounding effects. Including algorithms based on simple assumptions, such as phylogenetic placement [25], as baseline models for ML development should help in determining if machine learning is providing additional insights or simply recapitulating phylogenetic structure.

In conclusion, addressing the interplay between population structure and AMR prediction will require a multifaceted approach that includes improved sampling, algorithmic innovation, and systematic evaluation of proposed prediction methods. Only by confronting these challenges can we unlock the full potential of machine learning to provide actionable insights into AMR, advancing both our surveillance capabilities and our understanding of resistance mechanisms. These efforts will be critical to combating the global threat of AMR and ensuring the continued efficacy of life-saving antimicrobial therapies.

## Methods

### Data collection

Whole genome sequences and antibiotic phenotypes were extracted from [19] and [30]. 1,681 *E. coli* genomes from [19] were assembled from FASTQ files using Velvet (Version 1.2.10) [31] with a hash length of 45. Genome quality was checked using CheckM (version 1.2.1) [32] with the identical species name for the parameter "taxon_set species". Genomes with Completeness lower than 95% or Contamination higher than 5% were removed from subsequent analysis.

### Phylogenetic analysis

Genomes were annotated using Prokka (version 1.14.6) [33]. Annotation files were used as input for core genome alignment using Roary (version 3.7.0) [16] with a 95 minimum percentage identity for all genomes of each species. The core genome was defined once for each analyzed species. SNP-sites (version 2.5.1) [17] was used for SNP-calling on the core genome alignments generated by Roary. The phylogenetic tree was built on core genome alignments using IQ-TREE (version 2.2.0.3) [34] with the GTR model. Pairwise genome distance was calculated using the tree file from IQ-TREE and "phylogeny_distance.py" from Pyseer (version 1.3.10) [13] with the "--lmm" parameter. Sequence types were determined based on genes on https://pubmlst.org/data [35]. For *E. coli*, the #1 set of genes was used. For the visualization of the phylogenetic tree and related information, iTol (version 6.8) [36] was used. Only the top 25 sequence types with the highest number of genomes were shown in the iTol plots.

### Applying machine learning to predict antibiotic resistance

**Training target and model type.** LightGBM (version 3.3.2) [18] binary classification models were trained to predict resistant or susceptible phenotypes for each antibiotic of each species. Antibiotics with less than 1,000 genomes or with either resistant or susceptible samples higher than 80% were not included in the training.

**Training samples.** Genomes in each species were manually split into several clades based on the phylogenetic trees (S2–S6 Figs). For each antibiotic, clades with either resistant or susceptible samples less than 50 were excluded. Two schemes were tested. In scheme A (Fig 1C), one clade was selected to be paired with each of the other clades. If clade 1 contained sufficient resistant or susceptible samples, then it would be selected. Otherwise, other clades were tested for sufficient resistant or susceptible samples one by one until one clade was selected. Samples from the selected clade and one pairing clade were included for each training. Twenty resistant and 20 susceptible samples from the pairing clade were held-out for testing. Four scenarios were tested in scheme A. In each scenario, either all resistant or susceptible samples from one clade were excluded. When testing for including resistant and susceptible samples in training, between 10 and 90% of resistant samples and 90 and 10% of susceptible samples from one clade were included. In scheme B (Fig 1E), two scenarios were tested. In each scenario, either all resistant or susceptible samples from one clade were excluded, while both resistant and sensitive samples from all other clades were included. For all scenarios, 10-fold cross-validation was applied to resistant and susceptible samples in each clade and 90% of the samples were included in each fold according to the specific scenario in each scheme. In the random train-test split regime for both schemes A and B, training samples remained the same while the test set composition depended on the training exclusions. For example, if

resistant samples from clade N were excluded from training, only sensitive samples from clade N were included in the test set. For training models on each clade, all samples in the clade were included.

**Training features.** SNPs from SNP-sites and the presence and absence table from Roary were used as features. For the unitig analysis, unitigs were called using unitig-caller (version 1.3.1) from the Bifrost package [37] with the—call option to obtain a binary presence and absence table. Due to computational constraints, unitig-based analysis was performed on a subset of *K. pneumoniae* genomes (1,429 genomes from three representative clades out of the total 3,205). In each training, features with a variance equal to zero were removed.

**Hyperparameter tuning.** Hyperparameters for the LightGBM model were optimized using hyperopt (version 0.2.7) [38]. Search space included: 'max_bin' from range 50 to 500 with a step of 50, 'bagging_fraction' from range 0.01 to 1, 'bagging_freq' from range 0 to 10, 'feature_fraction' from range 0.01 to 1, 'subsample_for_bin' from range 30 to 0.8*sample size, 'max_depth' from range 1 to 16 with a step of 1, 'learning_rate' from range 0.01 to 1, 'lambda_l2' from range 0 to 100, 'min_data_in_leaf' from range 1 to 300, 'min_gain_to_split' from range 0 to 15, 'num_leaves' from range 2 to 100 with a step of 1. All samples were used for tuning. The mean of the balanced accuracy score in 10-fold cross-validation was used as the scorer. And the optimized set of hyperparameters was used in scenarios in the same scheme.

**Evaluation.** The held-out samples and the excluded samples were combined and an equal amount of resistant and susceptible samples were randomly drawn for calculation. The mean values of the scores over 100 repetitions were used.

## Applying machine learning to interpret model performance

The default random forest regression model from scikit-learn (version 1.0.2) [39] was used to predict the AUC scores of the models. Species, scenarios, clade-wise distance, sample size, and drug classes were included as features. Drug classes were defined according to CARD [40]. The clade-wise distance was calculated as the mean of all genome pair-wise distances between two clades. Species, scenarios, and drug classes were one-hot encoded. When training on scores from one species, the feature species was removed.

## Model interpretation

SHAP values were calculated using the 'shap_values' function in TreeExplainer from the Python package shap (version 0.39.0) [21] with all samples. The resulting SHAP values were visualized with the 'summary_plot' function provided in the same package.

## Supporting information

**S1 Data. The full list of tested antibiotics for each of the five species.** Column definitions: species: name of the species; antibiotic: name of the antibiotics; R: number of resistant genomes; S: number of susceptible genomes; size: Total number of genomes; MOA: mechanism of action for the antibiotic.
(XLSX)

**S2 Data. Model performance across all scheme scenarios.** Column definitions: AUC score; balanced accuracy score; Precision; Recall; species: name of the species that the model trained on; antibiotic: name of the antibiotic that the model trained on; MOA: mechanism of action for the antibiotic; case: testing scenario, corresponding to Fig 1. For example, A-a means scheme A training scenario a; R1: in scheme A, this denotes the number of resistant genomes in clade 1. In scheme B, it denotes the number of resistant genomes in the other clades where resistant or susceptible genomes were not excluded; S1: in scheme A, this denotes the number of susceptible genomes in clade 1. In scheme B, it denotes the number of susceptible genomes in the other clades where resistant or susceptible genomes were not excluded; Rn: in scheme A, this refers to the number of resistant genomes in the paired clades. In scheme B, it refers to the number of resistant genomes in the clade from which resistant or susceptible genomes were excluded; Sn: in scheme A, this refers

to the number of susceptible genomes in the paired clades. In scheme B, it refers to the number of susceptible genomes in the clade from which resistant or susceptible genomes were excluded; distance: clade-wise distance (See "Applying machine learning to interpret model performance" in Methods); model: training model type; scheme: training scheme; split: sampling methods used for calculating the scores are indicated. 'Scheme' refers to sampling according to the defined scheme scenarios, while 'Random' refers to random sampling. See Training samples in Methods for details; drug_class: drug classes according to CARD.
(XLSX)

**S3 Data. Excel workbook containing the source numerical data underlying figures in the manuscript.** Each worksheet corresponds to specific figures and includes all individual data points used to generate the displayed results. Column definitions: For sheets "Figs 1D, 1F, 1G, 2C, S7A and S10", "Figs 2B and S7BC (scheme A)", "Fig 2B (scheme B)", "S7BC Fig", and "S10 Fig": defined as in S2 Data. For sheets "Fig 2A (mean values)", "S2–S6C Figs (mean)", and "S9 Fig", shap_values: mean absolute SHAP values, features. For sheets "Fig 2A (SHAP values)" and "S2–S6C Fig (shap)", each column corresponds to a model feature. For sheets "S1A Fig", species; antibiotic: name of the species that the model trained on; clade; R: number of resistant genomes; S: number of susceptible genomes; size: Total number of genomes. For sheet "S1B Fig", columns are defined as in S1 Data. For sheet "S8 Fig", species; count: overlapping features in 50% summed SHAP features (%); top_N: the number of 50% summed SHAP features; count_top10: overlapping features in top 10 SHAP features (%).
(XLSX)

**S1 Fig. Overview of the collected data.** (A) The number of genomes included in the training for different antibiotics in five species. (B) The number of antibiotics in each MOA in five species. CWSI: cell wall synthesis inhibitor, DSI: DNA synthesis inhibitor, PSI: protein synthesis inhibitor, RSI: RNA synthesis inhibitor. The full list of tested antibiotics for each of the five species is included in S1 Data.
(TIFF)

**S2 Fig. Phylogenetic tree and model interpretation in *Escherichia coli*.** (A) Clade definition for model training, the antibiotic phenotypes, and the sequence types (ST) shown on the phylogenetic tree. S: susceptible, R: resistant. (B) The distribution of pairwise distances between genomes of clade 1 and other clades. (C) SHAP values for the top 10 features from a random forest model trained on AUC scores from both schemes A and B for *E. coli*. Underlying data are available in S3 Data and in the file S2_6B.tsv.gz on Mendeley Data under DOI: https://doi.org/10.17632/zs2mbjv7dn.3.
(TIFF)

**S3 Fig. Phylogenetic tree and model interpretation in *Klebsiella pneumoniae*.** (A) Clade definition for model training, the antibiotic phenotypes, and the sequence types (ST) shown on the phylogenetic tree. S: susceptible, R: resistant. (B) The distribution of pairwise distances between genomes of clade 1 and other clades. (C) SHAP values for the top 10 features from a random forest model trained on AUC scores from both schemes A and B for *K. pneumoniae*. Underlying data are available in S3 Data and in the file S2_6B.tsv.gz on Mendeley Data under DOI: https://doi.org/10.17632/zs2mbjv7dn.3.
(TIFF)

**S4 Fig. Phylogenetic tree and model interpretation in *Salmonella enterica*.** (A) Clade definition for model training, the antibiotic phenotypes, and the sequence types (ST) shown on the phylogenetic tree. S: susceptible, R: resistant. (B) The distribution of pairwise distances between genomes of clade 1 and other clades. (C) SHAP values for the top 10 features from a random forest model trained on AUC scores from both schemes A and B for *S. enterica*. Underlying data are available in S3 Data and in the file S2_6B.tsv.gz on Mendeley Data under DOI: https://doi.org/10.17632/zs2mbjv7dn.3.
(TIFF)

**S5 Fig. Phylogenetic tree and model interpretation in *Staphylococcus aureus*.** (A) Clade definition for model training, the antibiotic phenotypes, and the sequence types (ST) shown on the phylogenetic tree. S: susceptible, R: resistant. (B) The distribution of pairwise distances between genomes of clade 1 and other clades. (C) SHAP values for the top 10 features from a random forest model trained on AUC scores from both schemes A and B for *S. aureus*. Underlying data are available in S3 Data and in the file S2_6B.tsv.gz on Mendeley Data under DOI: https://doi.org/10.17632/zs2mbjv7dn.3. (TIFF)

**S6 Fig. Phylogenetic tree and model interpretation in *Streptococcus pneumoniae*.** (A) Clade definition for model training, the antibiotic phenotypes, and the sequence types (ST) shown on the phylogenetic tree. S: susceptible, R: resistant. (B) The distribution of pairwise distances between genomes of clade 1 and other clades. (C) SHAP values for the top 10 features from a random forest model trained on AUC scores from both schemes A and B for *S. pneumoniae*. Underlying data are available in S3 Data and in the file S2_6B.tsv.gz on Mendeley Data under DOI: https://doi.org/10.17632/zs2mbjv7dn.3. (TIFF)

**S7 Fig. Precision and recall scores for scheme A.** (A) Scores in each scenario for all antibiotics and species. (B) Precision scores and (C) recall scores when decreasing resistant samples and increasing susceptible samples from the paired clade were included in training, with susceptible samples considered as positives. Underlying data are available in S3 Data. (TIFF)

**S8 Fig. Overlapping predictive features across models trained on each clade.** (A) top 10 features and (B) top features with SHAP values summed up to 50% of total SHAP values. (C) The distribution of the number of features with SHAP values that summed up to 50% of total SHAP values. Underlying data are available in S3 Data. (TIFF)

**S9 Fig. Mean absolute SHAP values for features contributing to models trained on individual bacterial clades to predict ciprofloxacin resistance.** Bars indicate the average magnitude of each feature's contribution across all strains in the clade, with higher values reflecting greater influence on model predictions. SNPs are indicated by gene names followed by the notation (position, reference allele > alternate allele(s)), while presence/absence features are indicated by gene name alone. Plots are shown for (A) *Escherichia coli*, (B) *Salmonella enterica*, and (C) *Staphylococcus aureus*. Only clades with at least 50 resistant and 50 susceptible strains were included. Underlying data are available in S3 Data. (TIFF)

**S10 Fig. Comparison of different feature encoding strategies for predicting ciprofloxacin resistance in *Klebsiella pneumoniae*.** Boxplots show the distribution of the area under the ROC curve (AUC) scores for scheme A, evaluated across three clades. Models were trained using (i) presence/absence of unitigs, and (ii) the original feature set (SNPs from the core genome combined with accessory gene presence/absence). For comparison, models trained with the original SNP and accessory gene presence features were run both across all clades and across the same three clades used in the unitig analyses. Higher AUC indicates better performance. Underlying data are available in S3 Data. (TIFF)

## Author contributions

**Conceptualization:** Yanying Yu, Nicole E. Wheeler, Lars Barquist.

**Data curation:** Yanying Yu.

**Formal analysis:** Yanying Yu.

**Funding acquisition:** Lars Barquist.

**Investigation:** Yanying Yu, Lars Barquist.

**Methodology:** Yanying Yu.

**Project administration:** Lars Barquist.

**Resources:** Lars Barquist.

**Software:** Yanying Yu.

**Supervision:** Lars Barquist.

**Visualization:** Yanying Yu.

**Writing – original draft:** Yanying Yu, Lars Barquist.

**Writing – review & editing:** Yanying Yu, Nicole E. Wheeler, Lars Barquist.

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
