## [Editor Report · Decision Letter 0]

19 Feb 2025

Dear Dr Barquist,

Thank you for submitting your manuscript entitled "Biased sampling confounds machine learning prediction of antimicrobial resistance" for consideration as a Short Reports by PLOS Biology.

Your manuscript has now been evaluated by the PLOS Biology editorial staff, as well as by an academic editor with relevant expertise, and I am writing to let you know that we would like to send your submission out for external peer review.

Once your full submission is complete, your paper will undergo a series of checks in preparation for peer review. After your manuscript has passed the checks it will be sent out for review. To provide the metadata for your submission, please Login to Editorial Manager (https://www.editorialmanager.com/pbiology) within two working days, i.e. by Feb 21 2025 11:59PM.

Kind regards,

Melissa

Melissa Vázquez Hernández, PhD

Associate Editor

PLOS Biology

on behalf of

Richard

Richard Hodge, PhD

Senior Editor

PLOS Biology

rhodge@plos.org

---

## [Decision Letter · Decision Letter 1]

9 Apr 2025

Dear Dr Barquist,

Thank you for your patience while your manuscript "Biased sampling confounds machine learning prediction of antimicrobial resistance" was peer-reviewed at PLOS Biology. Please accept my sincere apologies for the delays that you have experienced during the peer review process. Your manuscript has now been evaluated by the PLOS Biology editors, an Academic Editor with relevant expertise, and by two independent reviewers.

In light of the reviews, which you will find at the end of this email, we would like to invite you to revise the work to thoroughly address the reviewers' reports.

As you will see, both reviewers are positive about your study and think it provides an interesting and cautionary message for the machine learning community. Reviewer #1 notes that the discussion of methods to control for phylogenetic structure should be expanded and how that information could be used in more complex ML setups. In addition, Reviewer #2 raises concerns that underlying biases in the hybrid methodological approach may not have been accounted for and could influence the findings. The reviewers requests additional discussions of the limitations or analyses.

Given the extent of revision needed, we cannot make a decision about publication until we have seen the revised manuscript and your response to the reviewers' comments. Your revised manuscript is likely to be sent for further evaluation by all or a subset of the reviewers.

**IMPORTANT - SUBMITTING YOUR REVISION**

*Re-submission Checklist*

*Published Peer Review*

*PLOS Data Policy*

*Blot and Gel Data Policy*

Best regards,

Richard

Richard Hodge, PhD

rhodge@plos.org

REVIEWS:

Reviewer #1: This manuscript tests the inaccuracy caused by phylogenetic dependency in machine learning-based detection of antibiotic resistance determinants. It is well known that taxa are not independent because they have traits that descend from common ancestors (this is also true for transferred traits, just the ancestry is different). The authors create cases where data is dependent in terms of phylogeny and show that predictors of antibiotic resistance are severely affected by this (even when positives are removed from clades). The phenomenon in itself is well known. But it seems the ML community is not very aware of it and this is a relevant cautionary tale.

Some major comments.

- The literature on the topic of phylogenetic inertia is actually very old (much older than the GWAS methods cited in the text) and it would be relevant to connect to it. It dates from at least Felsenstein, American Naturalist, 1985. A recent review is Dewar, Nature Rev Genetics, 2025.

- While the authors mention a few methods used in other contexts to control for phylogenetic structure, I found that part of the text to be superficial since it does not provide clear clues on how that could be used in more complex ML setups. Also, it seems to me important, in the spirit of the manuscript, to provide information on how one could make diagnostics that there is a problem on phylogenetic structure. This connects to the previous point and the existence of methods to assess phylogenetic dependence.

Some parts of the text need to be easier to read for the average reader of PLoS Biology.

- For example, the presentation of the dataset used is too brief in the main text.

- The meaning of SHAP values is not explained.

- "Based on the clade structure of our data" is too succinct, since one has to go deeper in the data to understand how clades are defined (and even then it's implicit that clades are actually sequence types).

- The results obtained about quinolonoes (difference between E. coli and S. aureus) are well known and published.

Reviewer #2: In this paper, the authors aimed to identify and quantify the contributions of various well-documented and common biases present in pathogen sample collections used for training machine learning classifiers for AMR prediction. The underlying datasets included five bacterial pathogens (both Gram-negative and Gram-positive) with matched AMR-associated metadata. The primary analysis was structured to exclude sensitive/resistant isolates from subpopulations/clades of the bacteria to identify the model's ability to identify generalisable predictors. This was further supported by a meta-model analysis of the resulting models to identify the most important predictive model features, alongside a series of clade-based analyses and further clade-based models to identify the top AMR predictors and the proportion of predictors that overlapped between analyses (i.e., were generalisable signals). Overall, the authors have produced a well-rounded and well-argued manuscript that has quantified and clarified a number of potential confounders previously observed and discussed by a range of authors. It also provides a useful framework for future researchers to appropriately evaluate work in this area.

The manuscript is well-structured, and the analyses are both interesting and timely, considering the growing interest in this research area. The authors have provided scripts and supporting information that would allow for the re-analysis of the data, with some exceptions (see below). This paper would generally be of interest to researchers in the fields of antimicrobial resistance prediction, machine learning, and machine learning applied to bacterial genomic datasets, where these kinds of confounding biases are both common and problematic. It might also be of interest to public health researchers who are focused on the development of these models for the monitoring and prediction of AMR resistance from public health genomics datasets.

Overall, I feel that the authors' claims are mostly well supported by the evidence they have provided, but they have not accounted for some of the underlying methodological biases that may have influenced their findings. This is a concern, considering that the paper purports to address and account for biases in sample collection but have not considered that they may have introduced methodological biases. I would like to see some further analysis and/or discussion of the limitations of their approach or the impact that it might have had on their results and subsequent conclusions prior to publication.

Major Points:

1/ The authors have opted to take a hybrid approach to generating genomic variants for downstream analysis by machine learning. This entailed the generation of core genomic variants (mapping and variant calling) and accessory genome content presence-absence information (pangenomic analysis). Whilst this approach does aid interpretation, it has some methodological downsides in that it introduces biases:

1a/ Gene presence-absence information relies on a pre-processing tool (Roary) providing the 'correct' clustering and classification of accessory genes. However, accessory genome content may be highly variable both in its genomic proportion per species (presence/absence) and in the underlying genetic variation present within accessory genes. Some of the species discussed within the paper have significantly larger and more diverse accessory genomes than others (e.g., E. coli, K. pneumoniae). As genetic variation 'within' accessory genes might be an important feature, it would not be identified using this analysis. This should be discussed in the context of the literature and the analysis.

1b/ The mapping/variant calling analysis of the core genome will be biased by the amount of sequence considered 'core' within and between species. This may be particularly egregious in Klebsiella and E. coli, which have particularly large accessory genomes relative to their core genomes. The results of SF2-6 suggest that between-clade distance is an important factor in the resulting meta-model/clade-based analysis. The authors have not sufficiently addressed whether this is due to real clade-specific differences or the proportion of shared core genomic variation between clades. As the authors have provided no information on the reference genomes used for mapping or how the core genome was generated (i.e., whether it was performed once for the whole species or separately between clades), it is difficult to identify if this would be a limitation of their approach. The first scenario would entail a strict core, where important predictive features may be absent from their dataset; the second would involve more or less input information for some clade comparisons.

2/ Further to the previous two points the authors might have instead considered instead taking an approach which is agnostic to bioinformatics pre-processing of their data, such as using k-mers or unitigs. I would like to see some discussion of the limitations of their methodological approach to genomic data in their discussion and perhaps some additional analyses of these potential biases on their outputs.

3/ The authors have not documented which reference genomes they used for the SNIPPY-based variant calling. Without this documentation, it would be impossible to reproduce their analyses.

Minor:

1/ "Indicating that the ML models were conflating indicators of lineage with genuine indicators of AMR." — I am not sure this is fully supported by the analysis at this stage of the paper. The assumption here is that the drop in precision/recall is due to the model predominantly learning features that are indicators of lineage rather than indicators of resistance. However, we would assume that AMR elements are not necessarily distributed evenly across clades and that certain clades might have more, fewer, or uneven distributions of particular AMR genes or variants. In light of this, the drop in the percentage of resistance might instead be considered as the model having less relevant/important information to learn from. Later figures and analyses cover and discuss this further (i.e., Supp F9). I would suggest moving such a strong conclusion to later in the paper, where it is better supported by the evidence.

2/ There are some general formatting issues with some of the existing figures, which reduce their interpretability and effectiveness. In particular, Figure 1 is difficult to read. The colours selected do not have sufficient contrast and do not display well. Additionally, the figure legend is not particularly clear in context.

3/ Figure 2: I am not sure if it is my screen but I literally couldn't see the bars in panels B+C. AUC 'socre' is incorrectly spelled.

4/ Figure S2-6 - Again, the choice of colours did not visualise well in panel B.

---

## [Decision Letter · Decision Letter 2]

20 Oct 2025

Dear Dr Barquist,

Thank you for your patience while we considered your revised manuscript "Biased sampling confounds machine learning prediction of antimicrobial resistance" for consideration as a Short Report at PLOS Biology. Your revised study has now been evaluated by the PLOS Biology editors, the Academic Editor and the original reviewers.

In light of the reviews, which you will find at the end of this email, we are pleased to offer you the opportunity to address the remaining points from the reviewers in a revision that we anticipate should not take you very long. We will then assess your revised manuscript and your response to the reviewers' comments with our Academic Editor aiming to avoid further rounds of peer-review, although we might need to consult with the reviewers, depending on the nature of the revisions.

In addition, I would be grateful if you could please address the following editorial and data-related requests that I have provided below (A-F):

(A) We routinely suggest changes to titles to ensure maximum accessibility for a broad, non-specialist readership. In this case, we would suggest a minor edit to the title, as follows. Please ensure you change both the manuscript file and the online submission system, as they need to match for final acceptance:

"Biased sampling driven by bacterial population structure confounds machine learning prediction of antimicrobial resistance”

(B) You may be aware of the PLOS Data Policy, which requires that all data be made available without restriction: http://journals.plos.org/plosbiology/s/data-availability. For more information, please also see this editorial: http://dx.doi.org/10.1371/journal.pbio.1001797

-Supplementary files (e.g., excel). Please ensure that all data files are uploaded as 'Supporting Information' and are invariably referred to (in the manuscript, figure legends, and the Description field when uploading your files) using the following format verbatim: S1 Data, S2 Data, etc. Multiple panels of a single or even several figures can be included as multiple sheets in one excel file that is saved using exactly the following convention: S1_Data.xlsx (using an underscore).

-Deposition in a publicly available repository. Please also provide the accession code or a reviewer link so that we may view your data before publication.

Figure 1D-G, 2A-C, S1A-B, S2B-C, S3B-C, S4B-C, S5B-C, S6B-C, S7A-C, S8A-C, S9A-C, S10

(C) Please also ensure that each of the relevant figure legends in your manuscript include information on *WHERE THE UNDERLYING DATA CAN BE FOUND*, and ensure your supplemental data file/s has a legend.

(D) Please note that we cannot accept sole deposition of code in GitHub, as this could be changed after publication. However, you can archive this version of your publicly available GitHub code to Zenodo. Once you do this, it will generate a DOI number, which you will need to provide in the Data Accessibility Statement (you are welcome to also provide the GitHub access information). See the process for doing this here: https://docs.github.com/en/repositories/archiving-a-github-repository/referencing-and-citing-content

(E) Please ensure that your Data Statement in the submission system accurately describes where your data can be found and is in final format, as it will be published as written there.

(F) Please ensure that you are using best practice for statistical reporting and data presentation. These are our guidelines https://journals.plos.org/plosbiology/s/best-practices-in-research-reporting#loc-statistical-reporting and a useful resource on data presentation https://journals.plos.org/plosbiology/article?id=10.1371/journal.pbio.1002128

- If you are reporting experiments where n ≤ 5, please plot each individual data point.

**IMPORTANT - SUBMITTING YOUR REVISION**

*Resubmission Checklist*

*Published Peer Review*

*PLOS Data Policy*

*Blot and Gel Data Policy*

Best regards,

Richard

Richard Hodge, PhD

rhodge@plos.org

REVIEWS:

Reviewer #1: I thank the authors for adequately tackling my comments and criticisms. I have no more remarks. The authors should check the text as there are some typos, like spaces after "-", and especially the references, several of which lack volume, page numbers and/or ms identifier. In my PDF the figures are of poor quality, bu maybe it's the site conversion to PDF that poses problem.

Reviewer #2: The paper has not changed substantially since the previous review, with the exception of the inclusion of analyses identifying potential confounding factors and a clearer explanation of the methodological approaches used within the manuscript.

Overall, I feel that the manuscript is well positioned within the current literature to provide a valuable contribution regarding the use of machine learning without accounting for phylogenetic or genetic relatedness — particularly in the prediction of phenotypes that may be unevenly distributed across phylogenies or between bacterial species. The additional analyses provided by the authors, alongside their expanded discussion and improved contextualisation within the broader literature, have strengthened this work.

While there remain aspects of this research that could be further explored, the study as presented is sufficiently complete to be of use to the research community. This is particularly relevant given recent publications claiming improvements in antibiotic resistance prediction, some of which may suffer from the same issues and considerations highlighted by the current authors. In this context, I believe this paper is both timely and may play a useful role in the current literature base.

Minor comments:

1/ There appears to be substantial variation in the data available for sensitive/resistant (S/R) isolates across classes (i.e. many unclassified or "white" isolates in SFig 2-6). It is not clear to me why this is the case. This variation is likely a characteristic of the underlying datasets themselves, which were in some cases collected by previous authors as benchmark datasets. However, I think there is a need to address the nature fo the datasetsm, the variable sample sizes and detail any impact this might have on the analyses performed across each split/ species/scenario and to consider how/if this variability might meaningfully influence the results.

2/ Figures S9 and S10: The figure legends are not sufficiently detailed to explain the content of these figures.

3/ I have installed and tested the scripts in the GitHub repo on a subset of the datasets provided. I feel the repo needs slightly more annotation and updated install instructions prior to publication. I had to additionally lightgbm and ipython to the conda env described in installation instructions to ensure the scripts function effectively. Otherwise the primary script completed on a subset (E. coli, amoxicillin)

4/ What is the 'shap' package/function referred to here - 'SHAP value plots were generated with the 'summary_plot' function in shap.'

---

## [Editor Report · Decision Letter 3]

13 Nov 2025

Dear Dr Barquist,

Thank you for the submission of your revised Short Report "Biased sampling driven by bacterial population structure confounds machine learning prediction of antimicrobial resistance" for publication in PLOS Biology. On behalf of my colleagues and the Academic Editor, Tobias Bollenbach, I am pleased to say that we can accept your manuscript for publication, provided you address any remaining formatting and reporting issues. These will be detailed in an email you should receive within 2-3 business days from our colleagues in the journal operations team; no action is required from you until then. Please note that we will not be able to formally accept your manuscript and schedule it for publication until you have completed any requested changes.

PRESS

Best wishes, 

Richard

Richard Hodge, PhD

rhodge@plos.org

PLOS
